# High Throughput Identification of the Potential Antioxidant Peptides in *Ophiocordyceps sinensis*

**DOI:** 10.3390/molecules27020438

**Published:** 2022-01-10

**Authors:** Xinxin Tong, Jinlin Guo

**Affiliations:** The Ministry of Education Key Laboratory of Standardization of Chinese Medicine, Key Laboratory of Systematic Research of Distinctive Chinese Medicine Resources in Southwest China, Resources Breeding Base of Co-Founded, College of Pharmacy, Chengdu University of Traditional Chinese Medicine, Chengdu 611137, China

**Keywords:** antioxidant peptide, high throughput screening, *Ophiocordyceps sinensis*

## Abstract

*Ophiocordyceps sinensi**s*, an ascomycete caterpillar fungus, has been used as a Traditional Chinese Medicine owing to its bioactive properties. However, until now the bio-active peptides have not been identified in this fungus. Here, the raw RNA sequences of three crucial growth stages of the artificially cultivated *O. sinensis* and the wild-grown mature fruit-body were aligned to the genome of *O. sinensis*. Both homology-based prediction and de novo-based prediction methods were used to identify 8541 putative antioxidant peptides (pAOPs). The expression profiles of the cultivated mature fruiting body were similar to those found in the wild specimens. The differential expression of 1008 pAOPs matched genes had the highest difference between ST and MF, suggesting that the pAOPs were primarily induced and play important roles in the process of the fruit-body maturation. Gene ontology analysis showed that most of pAOPs matched genes were enriched in terms of ‘cell redox homeostasis’, ‘response to oxidative stresses’, ‘catalase activity’, and ‘ integral component of cell membrane’. A total of 1655 pAOPs was identified in our protein-seqs, and some crucial pAOPs were selected, including catalase, peroxiredoxin, and SOD [Cu–Zn]. Our findings offer the first identification of the active peptide ingredients in *O. sinensis*, facilitating the discovery of anti-infectious bio-activity and the understanding of the roles of AOPs in fungal pathogenicity and the high-altitude adaptation in this medicinal fungus.

## 1. Introduction

*Ophiocordyceps sinensis* (Berk.), syn. *Cordyceps sinensis*, belongs to the family Ascomycetes and is a highly valued Traditional Chinese Medicine (TCM) fungus used in Asian countries for over 2000 years [1,2]. Its common name is the Chinese caterpillar fungus, and it naturally inhabits the alpine Qinghai–Tibetan Plateau in South East Asia with an altitude of 3000–5000 m above sea level [3]. Over 20 bioactive ingredients have been reported in this species, including adenosine, cordycepic acid, ergosterol, and polysaccharides, which are thought to be responsible for a range of health benefits, including anti-inflammatory, anti-tumor, immunomodulating, and antioxidative activities [4]. Bioactive peptides have been discovered from a range of prokaryotes and eukaryotes [5]. While bioactive peptides from fungi source account for a small proportion, mainly from *Trichoderma viride* [5]. To date, bioactive peptide compounds have not been identified in *O. sinensis.*

*O. sinensis* is an entomopathogenic fungus and has a complex life cycle. This parasitic fungus forms a unique complex with the larva of ghost moth caterpillars (*Thitarodes* spp.), and the larva progressively becomes stiff and coated with mycelia to mummify the larvae [6]. A small stroma bud emerges from the head of the sclerotium and forms the stalked fruiting body [6]. The immune response against the fungi would produce plenty of reactive oxygen species (ROS) to prevent pathogens [7]; furthermore, the fungi and the host itself would promote its ROS antioxidant defense system [7]. Omics studies showed that oxidoreductase putatively involved in the ecdysteroid metabolism of insect molting could have a potential relationship with the fungal pathogenicity in *O*. *sinensis* [8]. The sclerotial differentiation state in *Sclerotium rolfsii* concurred with increasing levels of lipid peroxides [9]. ROS initiates sclerotia formation and SOD gene expression increased with the development and maturation of the sclerotia [10].

*O. sinensis* has adapted to the harsh high latitude region, such as low temperature/oxygen/pressure and highly intensive ultraviolet (UV) light, where it occurs in nature. The ecological factors would promote the generation of ROS [8]. Peroxidase genes in *O*. *sinensis* remarkably expanded compared with other closely related fungal species that reside at low latitudes. Moreover, the change in the distribution of ROS caused by *SOD-1* mutation resulted in the loss of perithecial polarity initiated by light [11,12], indicating that ROS metabolism would be associated with sexual development [13]. It was suggested that ecological factors regulating the formation and development of the fruit-body might be mediated by ROS. Moreover, most genes related to fruit-body formation were enriched in terms of ‘oxidative stress’ and/or ‘osmotic response’ in *A. nidulans* [14]. Comparative transcriptome analysis of the three serial growth stages showed that the shared DEGs (differentially expressed genes) primarily were enriched in ‘the response to oxidative stress’ and ‘peroxidase activity’, indicating that the oxidation–reduction process would play important roles in the whole growth process [15]. Furthermore, ROS was found to play stage-specific roles in different growth stages in our previous study [16]. Therefore, to coordinate the balance of ROS at a non-toxic concentration, antioxidant proteins/peptides would be produced, such as SOD, catalase (CAT), peroxidase, mitochondrial peroxiredoxin PRX1, and peroxiredoxin, which have been found to be up-regulated during mummification and fruiting process.

In recent studies, a large number of antioxidant metabolites was found in *O. sinensis*, such as cordycepic acid, phenols, and vitamin B (riboflavin). Se-rich peptide fractions and bioactive peptides found in yeast could be promising antioxidants that can be used as a food additive to enhance health [17]. However, there are no reports of antioxidant peptides being identified for *O. sinensis*. Modern technologies including transcriptomics, genomics, and peptidomics are very useful tools to discover active peptides. Consequently, in this study, RNA-seqs and proteomic data of the serial and crucial growth stages of *O. sinensis* were sequenced and analyzed. Our study fills the gap of the lack of research on active peptide ingredients in *O. sinensis*, which will establish a platform for the development of antioxidants in this medicinal fungus.

## 2. Materials and Methods

### 2.1. Samples and Data Information

The samples of *O. sinensis* were collected from the artificial cultivation workshop at Chengdu University of Traditional Chinese Medicine, China. Larvae of the insect larva infested by *O. sinensis* were designated as the mycoparasite complex (IL). The mummified larvae coated with mycelia were designated as the sclerotium (ST). The samples of stroma with lengths < 1 cm were designated as the primordium (PR), and the fruiting body with mature ascus and ascospores was designated as mature fruit-body (MF). These samples were washed in 0.9% saline, immediately frozen in liquid nitrogen, and stored in −80 °C until use. The wild-grown *O. sinensis* (YF) with mature fruit-body were harvested at Aba Prefecture, Sichuan Province, China (4500 m, N31° 080 51.900, E102° 210 26.7800) in May 2019.

The RNA-seqs were downloaded from the National Center for Biotechnology Information (NCBI) Sequence Read Archive under the accessions GSE160504 (RNA-Seq).

### 2.2. Gene Annotation and Differential Expression Analysis

The *O. sinensis* genome was downloaded from the NCBI database (https://www.ncbi.nlm.nih.gov/assembly/GCA_012934285.1 (accessed on 20 March 2020)). Trimmed paired-end reads were aligned to the reference genome by using HISAT2 aligner (version 2.0.4) [18]. Reads were aligned to the genome from each sample and then assembled into transcripts by StringTie (version 1.3.4d) using default parameters [19,20]. To globally characterize the expression patterns of diverse RNA-Seq samples, paired-end reads were aligned back to the assembled transcripts using Bowtie 2.0 as the aligner [19]. Gene expression patterns were quantified using htseq-count (version: 0.11.3) based on the read numbers that were mapped to each gene. The mapped read numbers of each assembled transcript were estimated and were normalized by RESM-based algorithm to obtain fragments per kilobase of transcript per million mapped fragments (FPKM) values for each RNA-Seq sample using perl scripts in the Trinity package (v2.11.0) [20]. RSEM results of each replicate of the sample were merged as one matrix for downstream analyses.

Differentially expressed genes (DEGs) were identified by using the edgeR package (empirical analysis of digital gene expression in R), with a threshold of adjusted log2FC(log2fold-change) ≥ 1 and false discovered rate (FDR) < 0.05 as statistically significant. A Venn diagram was used to analyze the pDAPs distribution in different comparisons. The RNA expression patterns of the common pDAPs with *p*-value < 0.05 in the three comparisons (IL vs. ST, ST vs. MF, and MF vs. YF were further clustered by one-way hierarchical clustering implemented in hCluster R package (Euclidean distance, average linkage clustering). The Gene Ontology (GO) based annotation suite BLAST2GO v. 3.0 [21] was used to functionally annotate genes and assess the quality of our assembly. The transcripts were aligned against three public databases (NR, Swiss-Prot, and KEGG).

### 2.3. smORF Identification

The open reading frame (ORF) of each transcript was identified using Open Reading Frame Finder software (ORF finder, version: 0.4.1), set with ORF to at least 5 codes long, and beginning with any code. Then, ORFs with lengths ≤100 were retained and subsequently compared with the small opening reading frame (smORF) database (this database integrates the current reliable smORF databases, including sORFs (websites: http://sorfs.org/ (accessed on 10 July 2020), openprot https://openprot.org (accessed on 10 July 2020)), smprot http://bigdata.ibp.ac.cn/SmProt/ (accessed on 21 July 2020)), and refseq and proteins with length ≤100 in the Swiss-Prot database. Only the identity ≥ 30, query_coverage ≥ 30, and subject_coverage ≥ 30 were retained as the final smORF.

### 2.4. AOP Identification

Both homology-based prediction and de novo-based prediction methods were used to identify AOP [22]. The identification of AOPs were performed using BLAST searching against sequences deposited in AnOxPePred (http://services.bioinformatics.dtu.dk/service.php?AnOxPePred-1.0 (accessed on 10 September 2020)) [23], AOD, Aod-Pred (http://lin.uestc.edu.cn/server/AntioxiPred (accessed on 15 September 2020)), AOPs, and SVM databases (http://server.malab.cn/AOPs-SVM/index.jsp (accessed on 20 September 2020)) [24,25], with the threshold of identity ≥ 60% and query_coverage ≥ 80%. The de novo-based prediction was performed using AodPred software [22].

### 2.5. Protein Preparation

The samples were derived from the three growth stages (ST, PR, MF) of the cultivated *O. sinensis* and the wild specimens (YF). The protein preparation method was used according to a previous study [26]. Briefly, 0.1 g of each fresh sample was ground into a powder in liquid nitrogen, suspended with a solution with 100:1 (*w*/*v*) urea lysis buffer containing 8 M urea, 2 mM EDTA, 10 mM iodoacetamide (DTT), 25 mM iodoacetamide (IAA), and 1% protease inhibitor cocktail (Roche), and it was sonicated for 1 min. The protein in the supernatant was precipitated by adding 3 volumes of acetone at −20 °C for 2 h. After centrifugation at 12,000 rpm for 15 min at 4 °C, the protein pellet was collected and air dried. Protein was extracted by resuspending the dry pellet in UT buffer (8 M urea, 100 mM triethyl ammonium bicarbonate (TEAB)). Protein concentrations were determined using the Bradford protein assay (Bio-Rad, Hercules, CA, USA) according to the manufacturer’s instructions, and then samples were kept in −80 °C prior to use.

### 2.6. Peptide Isobaric Labeling

For each sample, 100 mg protein was reduced with 10 mM DTT for 1 h at 37 °C and then alkylated with 55 mM iodoacetamide (IAM) for 30 min in darkness. The pool protein was diluted by adding 100 mM TEAB to a final urea concentrations < 2 M and then digested with sequencing grade modified trypsin (Promega, Madison, WI, USA) at a ratio of 1:50 trypsin/protein (*w*/*w*) t for the first digestion at 37 °C overnight, and a ratio of 1:100 trypsin/protein (*w*/*w*) for a second 4 h digestion [27]. Subsequently, protein was desalted using a Strata X SPE column (Phenomenex Inc; Torrance, CA, USA) and vacuum-dried using a SpeedVac concentrator (Thermo, San Jose, CA, USA). Peptide was reconstituted in 20 mL 500 mM TEAB, processed according to the manufacturer’s protocol for an 8-plex iTRAQ kit (Sigma, Aldrich, St. Louis, MO, USA), and incubated for 2 h at room temperature. The labeled peptide was desalted and dried by vacuum centrifugation.

### 2.7. HPLC Fractionation and High-Resolution LC–MS/MS Analysis Based on Q Exactive

Briefly, the iTRAQ-labeled samples were reconstituted in 0.1% formic acid (FA), injected onto an Acclaim PepMap R 100 C18 reversed-phase pre-column (3 μm, 100 Å, 75 μm × 2 cm, Thermo Fisher Scientific, San Jose, CA, USA) at 5 μL/min in 100% solvent A (0.1% FA in water). Separation of peptides was performed using a reversed-phase column (Acclaim PepMap R RSLC C18, 2 μm, 100 Å, 50 μm × 15 cm) with an increase gradient of 0–8% solvent B (0.1% FA in 98% ACN) over 5 min, 8–25% B over 35 min, and 25–98% B for 10 min, and then kept in 98% for 8 min. The flow rate was kept constant at 300 nL/min on an ultimate 3000 system. The eluent was sprayed via an NSI source at an electrospray voltage of 2.5kV and then analyzed by MS/MS in Q Exactive HF(Thermo Fisher Scientific, San Jose, CA, USA) [28]. The mass spectrometer was operated in data-dependent mode by tandem mass spectrometry (MS/MS). Full-scan MS spectra (from m/z 350 to 1800) were acquired in the Orbitrap with a resolution of 70,000. MS data were obtained by selecting up to the 15-most-abundant precursors ions present in the survey scan (300–1800 m/z) for decision-tree-based ion trap higher-energy collisional dissociation (HCD) fragmentation [27]. The HCD collision energy was set at 32% and a dynamic exclusion duration of 10.0 s.

### 2.8. Data Processing

The MS/MS raw data were analyzed using the Sequest software integration in Proteome Discoverer (version 1.3, Thermo Scientific) and searched against *O. sinensis* (CGMCC 3.14243). Search parameters were as follows: trypsin as the cleavage enzyme, maximum of 2 missed cleavages, carbamidomethyl used as a fixed modification and oxidation (M), N-Term Acetylation and iTRAQ labeling used as fixed modifications [27]. The databank searches were performed using a peptide mass tolerance of 20 ppm and a product ion mass tolerance of 0.05 Da and an FDR cutoff 0.05. The pAOPs were searched against the peptides from our proteome data.

### 2.9. Statistical Analysis

Statistical analysis of relative gene expression levels were calculated via the 2^−ΔΔCt^ method, using R (V3.2) to estimate the correlations between the mRNA expression levels of 9 the AOPs determined through qRT-PCR [29]. The histograms were plotted using GraphPad Prism software 8.0 (GraphPad, La Jolla, CA, USA).

Proteome and transcriptome analyses were carried out on at least three independent biological repeats for each sample. Other experiments were performed using at least three independent biological repeats and for each biological repeat, at least three technical repeats were performed.

## 3. Results

### 3.1. Overview of Transcriptome and Differential Expression Analysis

Clean reads were obtained for each replicate of IL, ST, and MF, and aligned to the *O. sinensis* genome using HISAT2 [18], and assembled by StringTie, resulting in 65,534 exons and transcripts. For each replicate, over 20,694,375 reads were successfully mapped. The assembled results are shown in Appendix A. The efficiency of comparison with the reference genome was between 91.69% and 97.68% (Table 1). The correlation analysis of differential expression patterns revealed that the three biological samples in each group showed similar high performance, and the samples of the three stages could be clearly assigned to four clusters (Figure 1A). The IL and ST stages were found to group together, while the stages of YF and MF grouped together, indicating that the asexual hyphae stage (IF vs. ST) and the sexual stages (YF vs. MF) have more similar expression patterns than that in the comparison to ST and MF.

DEGs were calculated based on fragments per kilobase per million (FPKM), with an FDR of <0.05 and |log2(fold change, FC)| ≥ 1. This threshold resulted in a total of 1176 genes as significant DEGs in IF vs. ST (657 genes up-regulated, 519 genes down-regulated), 263 in MF vs. YF (503 genes up-regulated, 260 genes down-regulated), and 1916 in ST vs. MF (932 genes up-regulated, 984 genes down-regulated) (Figure 1B, Appendix A). Differential expression in our dataset is represented as a volcano plot (Figure 2). These results revealed that the expression profiles of the asexual and sexual growth stages (YF and MF) and (ST and IL) were more similar to each other than when compared across growth stages (ST and MF). Previous studies showed that the DEGs in the adjacent growth stages were enriched primarily with the response to oxidative stress [6]. In this study, we mainly focused on the AOPs that might be generated during the infection and sexual development process.

### 3.2. AOP Identification and Functional Annotation

A total of 607,300 ORFs was predicted by ORF finder (Appendix A). A total of 154,507 smORFs was identified by searching against databases. The length distribution of the smORF is shown in Figure 3A and Appendix A, showing that most of ORFs were distributed within 40–60 bp. The 607,300 ORFs were used to identify putative AOPs (pAOPs) using the homology-based prediction method against the UniProtKB/Swiss-Prot databases. The results showed that totals of 2276 pAOPs were identified by the homology-based prediction method and 6269 pAOPs were identified by the denovo-based prediction method. Four AOPs were shared by the two methods listed in Table 2. A total of 8541 AOPs was identified and listed in Appendix A. About 50% of the pAOPs contained more than 100 aa with an average length of 138 aa (Appendix A). The longest antioxidant peptide had 1870 aa and the shortest one had merely 9 aa (Appendix A). The length distribution of the pAOPs was shown in Figure 3B.

The GO function classifications of the pAOPs were also analyzed. The pAOPs were involved in a number of biological processes (BPs) that were typical for AOPs, including ‘cell redox homeostasis’ (19 AOPs), ‘response to oxidative stresses’ (8 AOPs), ‘ transport ’ (7 AOPs), ‘actin filament severing’ (5 AOPs), and ‘actin cytoskeleton organization’ (5 AOPs), as shown in Figure 4A and Appendix A. The pAOPs were involved in the cellular component (CC), including the terms ‘integral component of cell membrane’ (49 AOPs), ‘ribosome’ (12 AOPs), and ‘intracellular’ (10 AOPs) (Figure 4B, Appendix A). The DAPs were involved in molecular function (MF), including the terms ‘sequence-specific DNA binding’ (17 AOPs), ‘ATP binding’ (12 AOPs), ‘GTP binding’ (9 AOPs), ‘catalase activity’ (4 AOPs), ‘flavin adenine dinucleotide binding’ (FDD, 6 AOPs), ‘FMN binding’ (4 AOPs), and ‘metal ion transmembrane transporter activity’ (4 AOPs), as shown in Figure 4C and Appendix A.

### 3.3. The Differential Expression Analysis of AOPs

The identified AOPs were searched against the DEGs. A total of 1008 pAOPs was identified between stages (Figure 5, listed in Appendix A). Totals of 270 DAPs were detected MF vs. YF (178 up-regulated, 92 down-regulated), 415 DAPs in IL vs. ST (198 up-regulated, 217 down-regulated), and 682 DAPs in MF vs. ST (337 up-regulated, 345 down-regulated). The number of DAPs in ST vs. MF was the highest, while that in MF vs. YF was the smallest, indicating that the differences in DAP expression primarily occurred between the stages of ST and MF. The Venn map showed that there were 33 common DAPs in the three comparisons (Figure 5). Furthermore, RNAs profiles of the 33 DAPs were analyzed (Figure 6). The results showed that there were five clusters with visible difference expression patterns (Figure 6A–E) in the three comparisons. In cluster one and five, with six and two DAPs, respectively, there were lower expression levels in IL compared to in ST, and higher expression levels in ST compared to MF, indicating that they had a gradual increase during the serial growth stages from IL to MF and might be associated with the whole growth process. In cluster 2 and 3, with 15 and 5 pDAPs, respectively, there were higher expression levels in IL compared to ST and lower expression levels in ST compared to MF, a decrease expression profile and then an increase expression profile upon the transition from the asexual- to sexual- stage, illustrating that these pDAPs might play important roles in fungus–larva interaction and fruit-body maturation. In clusters three and five, with five and two pDAPs, respectively, there were lower expression levels in MF compared to YF, indicating that these DAPs might be associated with habitat-adaptation in this fungus. Combined with the results of the Heatmap (Figure 6F), we found that 9 DAP-matched genes were up-regulated in ST compared to the other stages, including MSTRG.2804, MSTRG.4043, MSTRG.5836, and MSTRG.1138; 5 DAPs matched genes were up-regulated in YF compared to the other stages, including MSTRG.2804, MSTRG.7447, MSTRG.6130, MSTRG.4471, and MSTRG.10768; and about 13 DAP-matched genes had higher expression levels in MF compared to the other growth stages, such as MSTRG.10214, MSTRG.4298, MSTRG.10541, and MSTRG.6418.

### 3.4. Validation by RT-PCR

To confirm the reliability of transcriptome analysis using Illumina sequencing, nine transcripts encoding pAOPs were selected for qPCR validation. Specific primers were designed as listed in Appendix A. Except for two genes (MSTRG.5870, MSTRG.10795), the other results of qPCR conformed to that of RNA-seqs (Figure 7), supporting the accuracy of the RNA-Seq and differential expression AOPs (DAPs) analysis.

### 3.5. iTRAQ Analysis and pAOPs Validation

The proteome of three growth stages of the cultivated *O. sinensis* were investigated. For visualization, PCA was performed. A total of 1414 proteins was identified using iTRAQ in these samples of different stages, and 1129 proteins were quantified. We detected 2343, 2511, 2972, and 2972 unique peptides; 3747, 3384, 2808, and 4080 unique peptides; 1725, 1830, 2210, and 3266 unique peptides; and 2921, 2960, 2892, and 3400 unique peptides in ST, PR, MF, and YF, respectively. The peptides identified are listed in Appendix A. In this study, the most spectra results had a mass error within ± 4 ppm, indicating that the mass spectrometer’s mass accuracy was normal (Figure 8A). The length of the peptides identified were distributed between 7 aa and 41 aa (Figure 8B), and 90% of the peptide length was distributed within 28 aa. MS data were deposited in iProX with the primary accession code PXD030687.

A total of 1655 pAOPs was found in our protein-seqs, shown in Appendix A. Combined with previous studies, some crucial pAOPs are listed in Table 3, including peroxiredoxin (PRX, MSTRG.2560.1.c1091-450, MSTRG.2560. 2.c1043-450, MSTRG.2560.3.c1039-446,IOZ07G2061.t1.c627-1), superoxide dismutase [Cu–Zn] (SOD, IOZ07G1895.t1.c465-1), and thioredoxin-like protein (IOZ07G1808.t1.1-651).

## 4. Discussion

*O. sinensis* has been reported to have various biological activities that are of relevance for development in pharmaceutical products. Recent studies have mainly focused on its sexual development and pharmacological studies [8]. Omics studies suggested that antioxidants would be largely generated during interactions between fungal pathogens and host insects and the fruit-body formation, as well as the high-altitude adaptation [16,30,31]. Here, using transcriptome and proteome data of three serial growth stages in cultivated *O. sinensis* and the wild-grown mature fruit-body, the putative AOPs were firstly predicted and analyzed. The correlation analysis of differential expression genes revealed that the three biological samples in each group showed a higher similarity, indicating that samples within each group had good repeatability. The expression patterns in IL vs. ST and YF vs. MF had a higher similarity compared to that in ST vs. MF, indicating that a higher difference occurred in the stages of asexual growth and sexual development. Here a combination of homology- and denovo-based prediction methods were used to identify AOPs. Homologous-based approaches used BLAST searching against sequences deposited in the present antioxidant peptides databases. As the present databases with antioxidant peptides are sparse, a denovo-based prediction method was also used to identify AOPs. A total of 8541 pAOPs was identified. Only four pAOPs were shared by the two analysis methods, indicating that most of the pAOPs have not yet been identified and needed to be verified, and the four pAOPs needed to be further studied.

GO analysis showed that 27 pAOPs enriched in biological processes were involved in ‘cell redox homeostasis’ and ‘response to oxidative stresses’. A total of seven pAOPs was enriched in the mature fruiting body for ‘catalase activity’. It is well known that AOPs are induced in response to oxidative stresses, the maintenance of cell homeostasis, and the involvement of large molecular antioxidants such as superoxide dismutase (SOD), catalase (CAT), glutathione peroxidase (GPx), and glutathione reductase (GR). There were five pAOPs that were found to be involved in ‘actin cytoskeleton reorganization’ and ‘actin filament severing’, respectively. Many cytoskeletal proteins are sensitive to ROS and are critical for the function of vascular cells, serving mechanical, organizational, and signaling roles [32], suggesting these AOPs might play important roles in redox regulation of actin cytoskeleton. In addition, most of the AOPs enriched in cellular components were involved in the term ‘integral component of membrane’. Four AOPs were located in ‘endoplasmic reticulum membrane (ERM)’. Furthermore, four AOPs were classified in the molecular function term of ‘metal ion transmembrane transporter activity’. A previous study showed that antioxidants reacting with ROS, RNS, and radicals were produced in association with electron transport in the endoplasmic reticulum [33], consistent with our results. Moreover, several AOPs were found to be involved in the biological processes of ‘transporter’ and ‘FAD/FMN binding’. With the response to oxidative stresses, mitochondria would produce a limited amount of ROS to initiate a molecular stress response by inducing defense enzymes such as SOD and catalase, as well as other stress defense pathways [34]. Thus, these AOPs might reduce oxidative stress via mitochondrial regulation. Moreover, AOPs would exert effective metal ion (Fe^2+^/Cu^2+^) chelating activity to defend againstoxidative stress.

The differential expression profiles of the 1008 pAOPs matched genes were analyzed. Similarly, the numbers of pDAPs in ST vs. MF was the highest, secondly in IL vs. ST, and the least in MF vs. YF, indicating that the differences primarily existed in different growth stages. Furthermore, the highly differentiated expressed pDAPs matched genes between stages were selected. The result showed that the highest difference existed in the comparisons of growth stages, suggesting that these pDAPs genes might be play crucial roles in the mummification of the infected larvae and fruit-body maturation. For example, the AOP matched catalase (MSTRG.8461) was found to be up-regulated in the highest level in ST compared to that in IL and MF. Catalase is one of the crucial antioxidant enzymes that mitigates oxidative stress to a considerable extent by destroying cellular H_2_O_2_ to produce H_2_O and O_2_. To avoid the infection of fungal pathogens, insect hosts often rapidly produce plenty of ROS to directly kill pathogen; the increase of catalase would detoxify ROS for host infection [8,35]. Most of oxygen-dependent oxidoreductases use FAD as a co-factor. A previous study showed that a large FAD-linked oxidase encoded by RSc0454 is required for pathogenicity [36]. In the present study, four AOPs were identified to match FAD linked oxidase (MSTRG.3794) and were remarkably up-regulated in ST compared to that in the other two stages, suggesting that the AOPs might be related to this fungal pathogenicity in *O. sinensis*. Moreover, three AOPs from MSTRG.10756, a zinc finger protein GIS2 OS, were remarkably up-regulated in MF compared to ST. C_2_H_2_-type zinc finger proteins (ZFPs) are thought to play important roles in modulating the responses of plants to drought, salinity, and oxidative stress, and different members have different roles during stresses. Moreover, the mutant of ste A, a transcription factor homeodomain C_2_H_2_ zinc finger protein was found to not form cleistothecia in *A. nidulans*, suggesting that these AOPs might be related to the adaptation to the high latitude and sexual development [37]. Serine/threonine-protein kinase SKY1(MSTRG.7638) has a much higher expression level in IL and MF compared to that in ST. MAP kinases belong to the family of serine/threonine protein kinases and are activated by a MAPKKK–MAPKK–MAP kinase cascade, which plays critical roles in pathogenicity and in fungal development [38]. It was suggested that the ROS signal and MAPK cascade might be cross-linked and crucial for the fungal pathogenicity and sexual development in *O. sinensis*. Sexual differentiation process protein isp7 (MSTRG.6754) has a much higher expression level in MF than in ST (Table 3) and could be related to sexual reproduction in this fungus. Generally, these differentially expressed antioxidant peptides/proteins would be induced by oxidative stresses during the periods of fungal–insect interaction and sexual development and participate in regulating fruit-body development, the mechanism of which need to be further studied.

A total of 1655 AOPs was identified in our proteomic data. For examples, alnexin (CNX)(MSTRG.2357) is an integral membrane protein of the endoplasmic reticulum, and it is also associated with mitochondria oxidative stresses [39,40]. Here, some AOPs were matched to a putative peroxiredoxin (MSTRG.2560, IOZ07G2061; Table 3). Peroxiredoxins are central elements of the antioxidant defense system and the dithiol-disulfide redox regulatory network of plant and cyanobacterial cells [41]. They serve in the context of photosynthesis and respiration, but also in metabolism and development of all tissues, in nodules as well as during seed and fruit development [42]. Peroxiredoxins were firstly identified in yeast [43]. Here, we firstly identified the putative AOPs matched to peroxiredoxin, which might serve in the fruit-body development and defense against oxidative stresses in *O. sinensis*. The AOPs (IOZ07G1895) matched to SOD [Cu–Zn], SOD, and CAT play a critical role in scavenging free radicals and therefore are used as indicators to evaluate the effect on the antioxidant defense system [44]. Previous studies revealed that SOD was involved in cold response and fruiting body development [10].

To conclude, the integration of transcriptome- and proteome-seqs to analyze the pAOPs in *O. sinensis* from this study pave the way for developing antioxidant peptides in this medicinal fungus and understanding the biology underlying fungal pathogenicity and the fruit-body development at high latitudes.

## Figures and Tables

**Figure 1 molecules-27-00438-f001:**
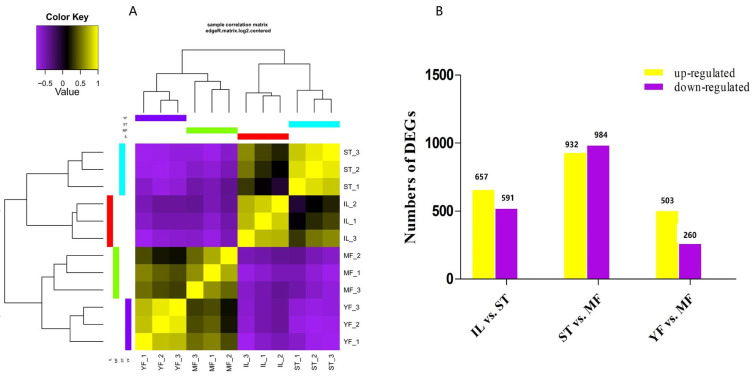
Heat map of the correlations between samples of the three adjacent growth stages of the cultivated *O. sinensis* and wild specimens in (**A**) and differential expression analysis of genes (DEGs) between growth stages in (**B**). YF is the wild-grown *O. sinensis*; IL is the mycoparasite complex of the cultivated *O. sinensis*; ST is the mummified larvae coated with mycelia, MF is mature fruiting body of artificial *O. sinensis*. The number of DEGs is shown on the top of histograms. Statistics of DEGs from *O. sinensis* between different growth stages.

**Figure 2 molecules-27-00438-f002:**
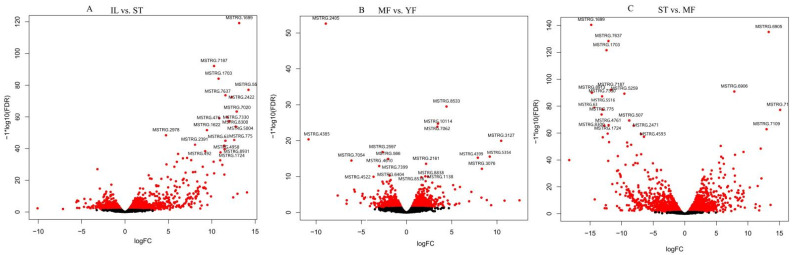
The volcano plots of differential expression genes (DEGs) between growth stages in *O. sinensis*. The X-axis represents fold change between IL and ST in (**A**), MF vs. YF in (**B**), and ST vs. MF in (**C**), while the Y-axis indicates significance of differential expression. The black dots signify no significant changes in the unigenes (|log2fold change| > 1, FDR < 0.05), while the red dots signify up- or down-regulated unigenes (|log2fold change∣ ≥ 1, FDR ≤ 0.05). IL is the mycoparasite complex of the cultivated *O. sinensis*; ST is the mummified larvae coated with mycelia of the cultivated *O. sinensis*; MF is mature fruiting body of the cultivated *O. sinensis*; YF is the wild-grown *O. sinensis*.

**Figure 3 molecules-27-00438-f003:**
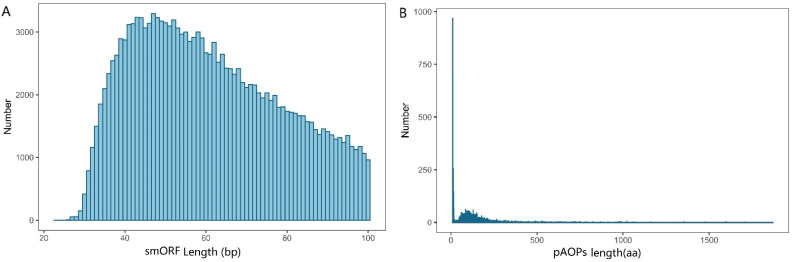
The length distribution of small open reading fragments (smORFs) in (**A**) and the putative antioxidant peptides (pAOPs) in (**B**).

**Figure 4 molecules-27-00438-f004:**
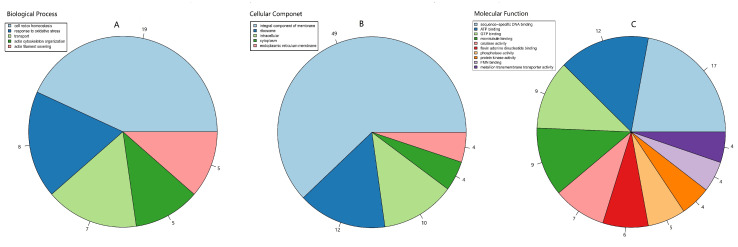
Pie chart of the top hit of GO function classifications of the AOPs in *O. sinensis*. The classification of biological processes (BP) in (**A**), cellular components (CC) in (**B**), molecular function (MF) in (**C**). GO slim categories from the Gene Ontology Consortium were used. Numbers following category name represent the number of the annotated AOP genes falling within that category from the top hit of GO annotation.

**Figure 5 molecules-27-00438-f005:**
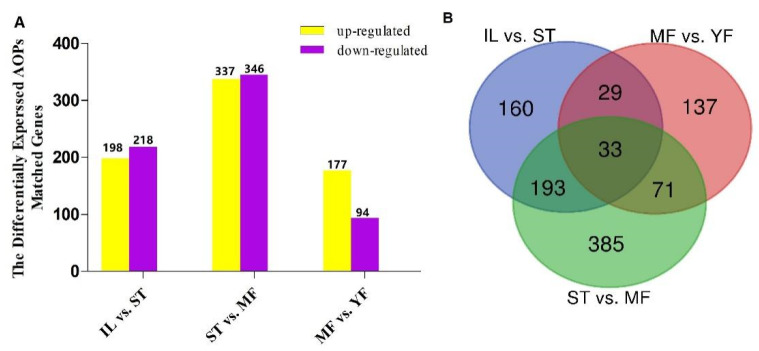
The differential expression analysis of AOPs (DAPs) between growth stages. (**A**) The number of DAPs was shown on the top of histograms. Statistics of DAPs in *O. sinensis* between different growth stages. (**B**) Venn diagram of DAPs comparing between different growth stages in *O. sinensis*. IL represents the mycoparasite complex of the cultivated *O. sinensis*; ST represents the mummified larvae coated with mycelia of the cultivated *O. sinensis*; MF represents the mature fruiting body of the cultivated *O. sinensis*; YF represents the wild-grown *O. sinensis*. The number of DAPs is shown on the top of histograms. Statistics of DAPs from *O. sinensis* between different growth stages.

**Figure 6 molecules-27-00438-f006:**
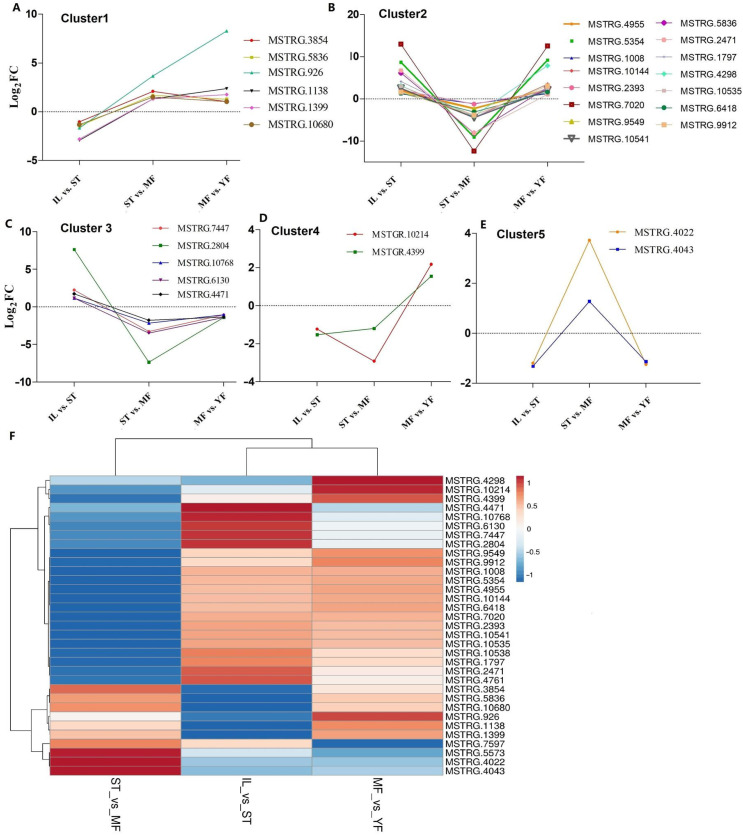
Clustering of the different expression patterns of 33 common putative antioxidant peptides (DAPs) matched genes in the three comparisons. (**A**–**E**) Five clusters with different expression patterns of the DAPs matched genes. (**F**) The heat map reveals relative abundance of the 33 DAPs matched genes in the comparisons. IL is the mycoparasite complex of the cultivated *O. sinensis*; ST is the mummified larvae coated with mycelia of the cultivated *O. sinensis*; MF is the mature fruiting body of the cultivated *O. sinensis*; YF is the wild-grown *O. sinensis*.

**Figure 7 molecules-27-00438-f007:**
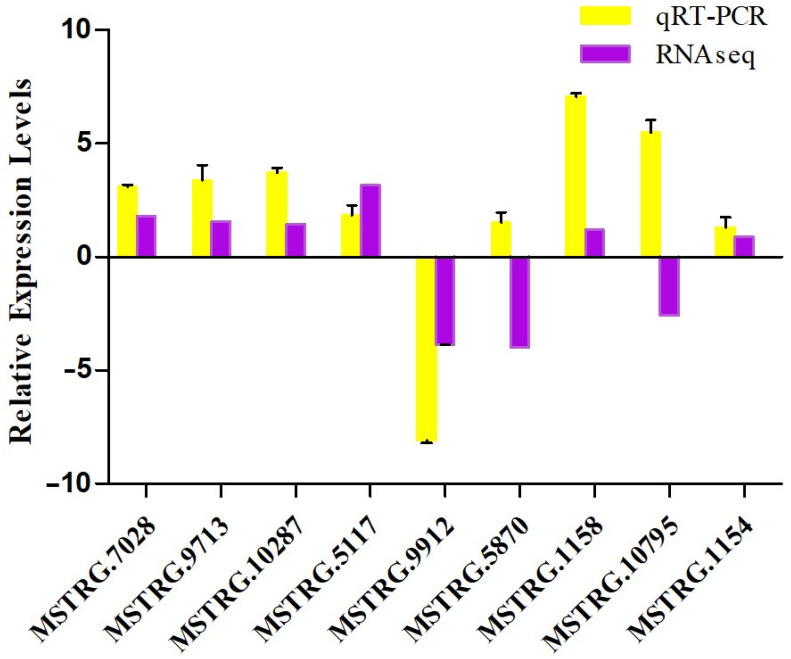
qRT−PCR validation of the pAOPs. The first four sets of bars represent the relative expression levels in each candidate AOPs identified in ST vs. IL (relative to IL), the middle three sets of bars represent that in ST vs. MF (relative to MF, and the latter three sets of bars represent that in MF vs. YF (relative to YF). Yellow bars represent qRT-PCR results (2^−ΔΔCt^). Error bars indicate the standard error. Purple bars represent the protein-seq results. 18sRNA was the internal reference. The X-axis represents the relative mRNA expression levels, and the Y-axis represents the AOPs gene ID. IL represents the mycoparasite complex of the cultivated *O. sinensis*; ST represents the mummified larvae coated with mycelia of the cultivated *O. sinensis*; MF represents the mature fruiting body of the cultivated *O. sinensis*; YF represents the wild-grown *O. sinensis*, and YF represents the wild-grown *O. sinensis* with mature fruit-body.

**Figure 8 molecules-27-00438-f008:**
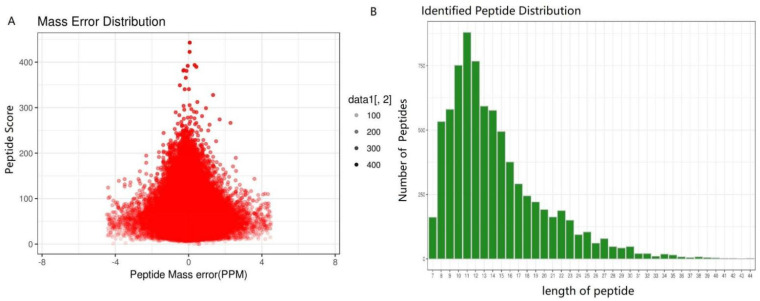
Proteins identified during the growth process of *O. sinensis*. (**A**) Mass error distribution of the identified peptides, (**B**) peptide length distribution.

**Table 1 molecules-27-00438-t001:** The read numbers and mapping rates of each sample. IL, the mycoparasite complex; ST, the mummified larvae coated with mycelia; MF, mature fruiting body of cultivated *O. sinensis*; YF, the wild-grown *O. sinensis* mature fruit-body.

Sample	Total_Reads	Overall_Align_Ratio
IL_1	23,401,999	94.09%
IL_2	23,360,595	94.07%
IL_3	22,210,722	93.87%
ST_1	25,374,934	92.90%
ST_2	28,656,031	93.10%
ST_3	21,814,291	92.56%
YF-1	22,893,233	95.53%
YF-2	32,326,044	95.45%
YF-3	20,694,375	95.09%
MF_1	23,239,296	96.88%
MF_2	22,212,619	91.69%
MF_3	22,963,667	97.68%

**Table 2 molecules-27-00438-t002:** The four AOPs identified by both of homology- and denovo-based prediction methods.

AOP Gene_Id	AOP Peptide
IOZ07G1895.t1.c465-1	LIGPHSVIGR
IOZ07G2719.t1.c853-821	GRDSTLGEIA
IOZ07G6427.t1.1-696	SYGVLLEDEGVALR
MSTRG.10682.1.c1440-1402	SGCLRPRANHLG

**Table 3 molecules-27-00438-t003:** Some critically annotated pAOPs in *O. sinensis* identified in our proteome data by searching against Swiss-prot with tophit.

The AOPs ID	Swiss-Prot ID	Annotation
MSTRG.8461	sp|Q92405|CATB_ASPFU	Catalase B OS
MSTRG.3794	sp|D7UQ40|SOL5_ALTSO	FAD linked oxidase
MSTRG.3929	sp|P38758|TDA3_YEAST	Putative oxidoreductase TDA3 OS
MSTRG.3935	sp|O74628|YQ53_SCHPO	Uncharacterized oxidoreductase C162.03 OS
MSTRG.7531	sp|Q9P7Q7|MAK1_SCHPO	Peroxide stress-activated histidine kinase mak1 OS
MSTRG.2357	sp|Q6Q487|CALX_ASPFU	Calnexin homolog OS
IOZ07G3010	sp|Q10058|YAM3_SCHPO	Putative oxidoreductase C1F5.03c OS
IOZ07G1808	sp|Q8TFM8|THIO_FUSCU	Thioredoxin-like protein OS
IOZ07G1895	sp|Q8J0N3|SODC_ISATE	Superoxide dismutase [Cu–Zn] OS
IOZ07G2061	sp|O43099|PMP20_ASPFU	Putative peroxiredoxin pmp20 OS
IOZ07G4162	sp|P34227|PRX1_YEAST	Mitochondrial peroxiredoxin PRX1 OS
IOZ07G6057	sp|P23710|NDUS3_NEUCR	NADH-ubiquinone oxidoreductase 30.4 kDa subunit, mitochondrial OS
IOZ07G6109	sp|Q9P7Q7|MAK1_SCHPO	Peroxide stress-activated histidine kinase mak1 OS

## Data Availability

The following information was supplied regarding data availability: Raw data of RNA-seqs are available in the National Center for Biotechnology Information (NCBI) Sequence Read Archive under the accessions GSE160504. Raw proteomics data was deposited in integrated proteome resources (iProX) with the primary accession code PXD030687. The *O. sinensis* strain is available in the China General Microbiological Culture Collection Center, accession number CGMCC 3.14243.

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
