# Peer review of "High Throughput Identification of the Potential Antioxidant Peptides in Ophiocordyceps sinensis"

_molecules, 2022, doi:10.3390/molecules27020438_

Round 1
Reviewer 1 Report
v
Dear Editor, dear Authors,
I have carefully read the manuscript and find it very insightful and significant for the scientific community. I found that the topic is very interesting, the paper is straightforward, well-written and well structured, the available scientific data are summarized and discussed clearly. In general, the paper clearly presents the research done. I suggest that the paper be accepted after some minor corrections listed bellow.
Figure 2 – the marks on the volcano plots are not visible. It should be larger.
Figure 4 – the marks on the legend are not visible. Letters in the legend should be larger, also the numbers on the pie charts.
Authors should check if the other graphics are legible.
Author Response
- The authors use cordyceps acid and cordycepic acid. Please standardize if they are the same compounds.
Answer: ‘cordyceps acid’ was revised to ‘cordycepic acid’.
- Section 2.7. Authors wrote "the iTRAQ-labeled samples were reconstituted in 0.1% formamide (FA)" Did you really use formamide? I guess you mean formic acid.
Answer: yes, it was revised to ‘formic acid’.
- Section 2.7 Authors used two additives in LCMS solvent system. FA in ACN and acetic acid in water. Can you explain the reason?
Answer: I checked this method. Some content should be revised. In fact, when inject sample onto the pre-column, 0.1% formic acid (FA) in water was used rather than acetic aid in water. And then the peptides were separated with solvent B ( FA in ACN ) in gradient. So this section should be revised as follow. The revised content was marked in red. Besides, ‘5 mL/min in 100% solvent A (0.1 M acetic acid in water)’ in front of the word marked in yellow was deleted.
Briefly, the iTRAQ-labeled samples were reconstituted in 0.1% formic acid (FA), injected onto an Acclaim PepMap R 100 C18 reversed-phase pre-column ( 3 μm, 100 Å, 75 μm × 20cm, Thermo Fisher Scientific, San Joes, USA ) at 5 mL/min in 100% solvent A (0.1% FA in water). Separation of peptides was performed using a reversed-phase column (Acclaim PepMap R RSLC C18, 2μm, 100 Å, 50 μm × 15cm) with an increase gradient from 0- 8% solvent B (0.1% FA in 98% ACN) over 5 min, 8 -25 % B over 35 min, 25- 98% B for 10 min and kept in 98% in 8 min. The flow rate was kept constant at 300 nL/min on an ultimate 3000 system. The eluent was sprayed via NSI source at the 2.5 kV electrospray voltage and then analyzed by MS/MS in Q Exactive HF(Thermo Fisher Scientific, San Joes, USA) [28].
- Section 2.7 Description of column used seems to be incorrect: "(Ac- 170 claim PepMap R RSLC 139 C18, 2 mm, 100 Å, 50 mm × 15 cm). Please specify the particle size in μm. The column dimension (50 mm × 15 cm) seems to be incorrect. Please specify in "mm"
Answer:yes, the specification should be revised to ‘PepMap R RSLC 139 C18, 2 μm, 100 Å, 50 μm × 15 cm)’.
- It would be worth to add some screens (Supplementary data) of peptides or proteins identified in bioinformatic analysis (e.g. ProteomeDiscover)
Answer:yes, I added the peptides identified through bioinformatic analysis in Table S10. Please check.
- Please also pay attention once again to the correct formatting of the references
Answer: yes, I revised the references format. Please check in this version of manuscript.

Reviewer 2 Report
The presented study aimed to identify bioactive peptide in Ophiocordyceps sinensis. The topic seems to be interesting due to the application of these fungi in alternative medicine, I recommend to publish this manuscript after minor revision. After careful analysis of this article I have some comments:
- The authors use cordyceps acid and cordycepic acid. Please standardize if they are the same compounds.
- Section 2.7. Authors wrote "the iTRAQ-labeled samples were reconstituted in 0.1% formamide (FA)" Did you really use formamide? I guess you mean formic acid.
- Section 2.7 Authors used two additives in LCMS solvent system. FA in ACN and acetic acid in water. Can you explain the reason?
- Section 2.7 Description of column used seems to be incorrect: "(Ac- 170 claim PepMap R RSLC 139 C18, 2 mm, 100 Å, 50 mm × 15 cm). Please specify the particle size in μm. The column dimmension (50 mm × 15 cm) seems to be incorrect. Please specify in "mm"
- It would be worth to add some screens (Supplementary data) of peptides or proteins identified in bioinformatic analysis (e.g. ProteomeDiscover)
- Please also pay attention once again to the correct formatting of the references
